# The Role of Macrophages in Vascular Repair and Regeneration after Ischemic Injury

**DOI:** 10.3390/ijms21176328

**Published:** 2020-08-31

**Authors:** Huiling Hong, Xiao Yu Tian

**Affiliations:** CUHK Shenzhen Research Institute, School of Biomedical Sciences, Heart and Vascular Institute, Chinese University of Hong Kong, Hong Kong, China; Flora_Hong@link.cuhk.edu.hk

**Keywords:** macrophage, angiogenesis, peripheral arterial disease, M1/M2, vascular repair

## Abstract

Macrophage is one of the important players in immune response which perform many different functions during tissue injury, repair, and regeneration. Studies using animal models of cardiovascular diseases have provided a clear picture describing the effect of macrophages and their phenotype during injury and regeneration of various vascular beds. Many data have been generated to demonstrate that macrophages secrete many important factors including cytokines and growth factors to regulate angiogenesis and arteriogenesis, acting directly or indirectly on the vascular cells. Different subsets of macrophages may participate at different stages of vascular repair. Recent findings also suggest a direct interaction between macrophages and other cell types during the generation and repair of vasculature. In this short review, we focused our discussion on how macrophages adapt to the surrounding microenvironment and their potential interaction with other cells, in the context of vascular repair supported by evidences mostly from studies using hindlimb ischemia as a model for studying post-ischemic vascular repair.

## 1. Introduction

Tissue repair and regeneration are the evolutionarily conserved protective processes against injury in living organisms [1]. In response to various types of injuries, whether it is toxic or mechanical, three well-recognized overlapping stages take place including the inflammatory responses, the proliferative phase, and tissue remodeling [2]. Immediately following injury, hemostasis and coagulation occur with the expression of chemokines and adhesion molecules on the injured endothelium, resulting in vascular hyper-permeability, infiltration of leukocytes, and later the disruption of the basement membrane which further facilitates the infiltration of leucocytes into the injured sites [3] The recruited neutrophils, monocyte-derived macrophages (MoM), and the tissue-resident macrophages (TrM) contribute to the phagocytosis of necrotic cells or pathogens at the injured tissue at different stages in the purpose to eliminate the insult. Subsequently, the proliferative phase is initiated to restore tissue integrity and function [4]. The proliferation of structural cells and angiogenesis to restore blood supply are critical events during the regenerative phase. At the same time, immune cells particularly macrophages continue to respond to the microenvironment and as a result, they release factors timely to facilitate remodeling and regeneration.

Studies using animal models of cardiovascular diseases have provided a clear picture of the effect of macrophages and their phenotype during injury and regeneration. The role of macrophages upon injury in different organs are reviewed previously [5,6]. Although damage to blood vessels is commonly associated with diverse types of tissue injury, in this short review, we focus our discussion on how macrophages adapt to the surrounding microenvironment and their potential interaction with other cells, in the context of vascular repair supported by evidence from both mouse and human studies.

## 2. The Role of Macrophages in Vascular Repair and Regeneration

The importance of macrophages in vascular repair has been implied to be involved in various injuries, such as peripheral artery diseases (PAD), myocardial infarction, and stroke. During vascular injury and repair, macrophages sense the vascular injury and accumulate at the site of injury, contributing to the neovascularization and promoting the recovery of blood flow. Recent studies on macrophages and angiogenesis provide mechanistic insights into the contribution of macrophages in vascular repair. The functions of macrophages in tissue and vessel repair include (i) secretion of pro-inflammatory cytokines and chemokines to maintain initial leukocytes infiltration, (ii) removal of invading pathogen and necrotic cell debris through phagocytosis, (iii) releasing matrix metalloproteinase (MMP) for extracellular matrix remodeling, (iv) promoting angiogenesis through guiding the sprouting of new blood vessels and stimulating the proliferation of endothelial cells (EC) and smooth muscle cells (SMC) [7]. At the molecular level, several key regulators related to macrophage function or produced by macrophages have been identified to participate in the restoration of blood perfusion, as summarized in Table 1.

### 2.1. Influence of Macrophage Polarization on Vascular Inflammation and Repair

Both TrMs and MoMs are highly plastic and adaptive in response to local stimuli. They are capable of acquiring very distinct functional phenotypes in response to stimuli at different times during disease development. Such phenotypical changes have important implications in many conditions, such as cancer, chronic inflammation, tissue injury, and repair. During tissue injury and repair, initially macrophages usually acquire the highly pro-inflammatory phenotype which is often referred to as classically activated macrophages (M1), due to the fact that they are often induced by T helper cell type 1 (Th1) cytokines and bacterial wall components, during its initial discovery. During the later stage, there is a substantial change of macrophage phenotype to the anti-inflammatory alternatively activated macrophages (M2), known to be triggered by T helper cell type 2 (Th2) cytokines. There is also a subtype characterization of M2 macrophages, including M2a, M2c, M2f, etc., based on the stimuli, which could be interleukins IL-4, IL-13, IL-10, transforming growth factor beta (TGF-β), apoptotic cells, etc. Biology of macrophage phenotype switch and polarization has been reviewed elsewhere thoroughly [1,28].

This phenotypical change between M1 and M2 macrophages is also important during post-ischemic vascular injury and repair. In general, the M1 macrophages are the predominant subset during the inflammatory phase while the M2 macrophages function to be pro-angiogenic and to enhance wound healing at the later stages [29]]. Within several hours post-injury, M1 macrophages are induced by pathogen-associated molecular patterns (PAMPs) or damage-associated molecular patterns (DAMPs). The activation of macrophages also facilitates the disruption of endothelium tight junction and the upregulation of EC adhesion to facilitate recruitment of neutrophils and monocytes. Upon the removal of harmful stimuli by neutrophils and macrophages, induction of angiogenesis mediated by M2 macrophages is indispensable for inflammation resolution and vascular repair [30]. Both M2a induced by IL-4 and M2c induced by IL-10 are capable of enhancing angiogenesis in vivo as demonstrated by matrigel plug assay. M2a expresses more fibroblast growth factor (Fgf2), C-C Motif Chemokine Ligand (Ccl2), and insulin-like growth factor-1 (Igf1) to stimulate angiogenesis, whereas M2c expresses more placental growth factor (Pgf) [31]. A later study confirmed that the delivery of M2a and M2c macrophages in vivo enhances post-ischemic angiogenesis and increases reperfusion recovery [32]. Interestingly, under hyperglycemia or hyperlipidemia, two common risk factors of PAD, there is an epigenetic alteration leading to hypomethylation of the promoters of many pro-inflammatory M1 genes and hypermethylation of promoters of many anti-inflammatory and pro-angiogenic M2 genes, which contributes to the impaired vascular repair [33]. However, other evidences also suggested that M1 macrophages produce more vascular endothelial growth factor (VEGF), FGF2, and IL-8, all of which are pro-angiogenic and stimulate cell migration, although M1 also produces several highly pro-inflammatory cytokines such as IL-1β and TNFα [34].

Apart from the common pro-angiogenic growth factors and chemokines such as VEGF, PGF, and IL-8 which could be produced by more than one macrophage subsets, many other molecules are preferentially produced by one specific macrophage subtype. Early work showed that M2 macrophages induced by IL-4 and also neutrophils express the highest level of pro-MMP9 which stimulates angiogenesis in vitro [35]. Importantly, M2 macrophages produce less TIMP1 which is the enzyme forming a complex with MMP9 to reduce its activity [35], indicating MMP9 is one of the factors contributing to the pro-angiogenic activity of M2 macrophages. In addition, M2a produces more platelet derived growth factor-BB (PDGF-BB), which is required to recruit pericytes, and M2c also produces more MMP9 [34]. A recent study also showed that in bioengineered vascular grafts, macrophages of distinct phenotypes can modulate the EC phenotype and behavior [36]. M1 induced by lipopolysaccharide or interferon gamma (IFNγ) and M2c induced by IL-10 are more involved in cell migration and the emergence of Tip cells in sprouting angiogenesis, whereas M2a induced by IL-4, and M2f induced by the apoptotic body are more involved in proliferation, pericyte and SMC differentiation for vessel tube formation [36]. M2f also shows a unique correlation with endothelial integrity, suggesting a role in stabilizing functional vessels, whereas M2c at some point suppresses sprouting angiogenesis and branching [36]. These studies indicated that different macrophage subsets participate in vascular repair probably at different stages and on different aspects of vascular repair.

Many other factors can modulate macrophage polarization and may also be involved in vascular repair. A small molecule, named itaconic acid or itaconate, which is a metabolite produced by macrophage through the tricarboxylic acid (TCA) cycle, is known to directly target succinate dehydrogenase induced oxidation of succinate, thus modulating macrophage metabolism by suppressing O_2_ consumption and therefore confining cytokine production [37]. Reduction of itaconic acid production, due to microRNA miR-93 induced inhibition of its synthesizing enzyme immune-responsive gene 1 protein (IRG1), induces M2-like polarization and improves angiogenesis and vascular perfusion after hindlimb ischemia (HLI) in mice [24]. This finding provides insights into the metabolic adaption of macrophages during the tissue repair process.

One of the most well-studied M2 marker Arginase 1 (encoded by *Arg1*) may also be functionally important in the M2 phenotype involved in vascular repair. Although global homozygous Arg1 deletion is lethal, one study showed that in heterozygous *Arg1* knockout mice, retinal inflammation, neuronal degeneration, and microvascular dysfunction are enhanced after ischemic reperfusion injury [38]. The same study also showed that Arg1 from macrophages rather than ECs is responsible for the worsened injury. In addition, Arg1 in a PEGylated form (conjugated with polyethylene glycol) can be delivered in vivo which dampens the inflammatory responses, indicating that Arg1 could be considered as a potential new approach for treatment of inflammatory diseases [38]. Arg1 could also be used as a target and marker of M1-M2 switch. For instance, one study using human placenta derived mesenchymal stem cells (MSCs) delivery in vivo to induce angiogenesis in mice after HLI found that these MSCs can induce an Arginase-1 producing M2-like macrophage subset, therefore enhancing vascular repair [39]. Exosomes produced by adipose-derived stem cells are also capable of polarizing macrophage to a M2 phenotype, with high expression of Arg1 and mannose receptor, therefore promoting angiogenesis both in vitro in a matrigel plug assay, as well as in vivo in mouse HLI model [40]. In summary, modulating the macrophage phenotypes can be considered both as an outcome and also as a mediator for developing new therapies for PAD. However, the sophisticated microenvironment and the spatio-temporal regulation of vascular repair by distinct macrophage subtypes also generate hurdles for potential therapies.

### 2.2. Chemotaxis and Recruitment of Various Cell Types during Vascular Repair

Restoration of perfusion involves both angiogenesis and arteriogenesis which is the collateral artery growth. The recruitment of new ECs, as well as blood monocytes, and macrophages may be involved in arteriogenesis. Early work using *Ccr2* knockout mice showed that without the CCL2-CCR2 (C-C motif chemokine receptor 2) chemotaxis, perfusion recovery is delayed, resulting from a smaller collateral artery with fewer macrophages in the perivascular spaces from these vessels. This indicates an important role of CCR2 dependent recruitment of circulating monocytes to the injury site to help with arteriogenesis [9,10]. Similarly, a later study using the skin wound model also revealed that the CCR2-dependent recruitment of CCR2^+^Ly6c^+^ monocytes into the wound is critical for wound healing due to the VEGF produced by these cells particularly during the early phase after injury [10]. These VEGF-producing MoMs could be either M1- or M2-like phenotypes. VEGF produced by macrophages contributes mostly to the sprouting angiogenesis while VEGF produced by epidermal cells contributes more to vessel maturation at a later stage [10]. Many factors are involved in post-ischemic angiogenesis through CCL2. For example, PGC1α, which is a transcriptional regulator of oxidative metabolism, also stimulates angiogenesis and facilitates the formation of mature functional vessels partially through VEGF [12]. Interestingly, PGC1α also induces secreted phosphoprotein 1 (SPP-1) synthesis and secretion from myocytes which stimulates macrophages to induce CCL2, and further activates ECs, pericytes and SMCs. Without SPP-1, PGC1α-induced vessel formation is aberrant and leaky [12]. On the contrary, another member of the phosphoproteins, the vasodilator-stimulated phosphoprotein (VASP), produced from mononuclear cells, can form complex with CCR2 and β-arrestin 2, leading to reduced infiltration of leukocytes into the injury site after ischemia [15]. Blockage of CCR2 also blunts the increased leukocyte infiltration in *Vasp* knockout mice and enhances perfusion recovery [15]. Importantly, hyperglycemia related high expression of advanced glycation end products (AGE) and their receptors (RAGE, encoded by *Ager*) enhances the CCL2 dependent recruitment of pro-inflammatory monocytes/macrophages resulting in excessive macrophage infiltration, causing impaired angiogenesis and delayed flow recovery [8], which might be one of the mechanisms leading to delayed vascular repair in diabetes.

Apart from CCR2-dependent chemotaxis, several other factors involved in chemoattraction, which may or may not involve CCR2, have also been identified (Figure 1). Other factors involved in the recruitment of differentiation of monocytes, for example granulocyte macrophage colony-stimulating factor (GM-CSF) and macrophage colony-stimulating factor (M-CSF), may also help to enhance the efficacy of bone marrow-derived mononuclear cells to promote vascular perfusion recovery, which may involve Foxp3 dependent regulatory T cells to modulate angiogenesis [41]. In the brain, one subset of microglia expressing neurophilin-1 (encoded by *Nrp1*) is mobilized after vascular injury and these cells respond to Semaphorin 3A as well as VEGF, to induce active proliferation of ECs and microvascular sprouting [13]. This mechanism is important for pathological neovascularization in retinopathy, but whether a similar mechanism works in other peripheral tissues, such as muscle or heart, is not fully understood. Another CXC chemokine receptor CXCR3, which uses CXCL10 as the ligand, is also involved in vascular repair. *Cxcr3* deficient mice showed delayed perfusion recovery and less capillary angiogenesis accompanied by less VEGF production [11]. Interestingly, CXCR3 is also involved in T cell attraction [11], suggesting that CD4^+^ T cells may also be involved in vascular repair through mechanisms reviewed elsewhere [42].

### 2.3. Phagocytic Macrophages in Vascular Repair

Phagocytosis is one of the most compelling features of macrophages. Early activation of macrophages by DAMPs or PAMPs induces the expression of the IFNγ in macrophages, and polarizes macrophage to a M1 phenotype to ingest the invading organisms and necrotic stromal cells. The infiltrated neutrophil is also a major player to perform phagocytosis. Upon the removal of the harmful stimuli, neutrophils undergo apoptosis and are cleared by M2 macrophages along with the induction of CD163, CD206, and TGFβ1 [29]. Although the IFNγ-mediated phagocytosis is necessary for tissue and vascular repair, persistent inflammation also exacerbates tissue injury. In the diabetic mice with enhanced IFNγ signaling, the protective effects of microglia against brain microbleeds are impaired under hyperglycemia while inhibition of IFNγ restores the reparative function of microglia to vessel damage [43]. Phagocytosis of apoptotic neutrophils by macrophages terminates the inflammatory stage and initiates the reparative process [29]. Signaling pathways involved in the phenotypic switch of phagocytic macrophage include AMPK, p38/MKP1, and CREB-C/EBPβ pathways, which have been demonstrated in the skeletal muscle regeneration [44,45,46]. However, the upstream cues for the timing control of macrophage polarization to enter repair stages, especially in vascular repair, are not fully characterized.

### 2.4. Pro-Angiogenesis Signaling Networks between Macrophages and Endothelial Cells

Injury of blood vessels is a series of events including the disruption of blood flow, cell damage, and necrosis. Disruption of blood flow results in the reduction of oxygen and nutrients supply. Such a hypoxic environment-triggering stabilization of hypoxia-inducible factor (HIF) activates a pro-angiogenic signaling network in different cell populations [47]. It is not surprising that global knockout of HIF-1α impairs vascular development and repair in zebrafish [48]. The macrophages are the selective responder of damage-associated hypoxia, as evidenced by the fact that the majority of hypoxic cells are macrophages at the site of neovascularization upon injury [49]. Intramuscular injection of HIF-1α-activating monocytes improves blood flow after femoral arterial injury [50].

The upregulated HIF protein induces two ligand-receptor systems in macrophages and ECs, including the VEGF-VEGFR and the angiopoietins-Tie receptors (ANG-TIE2) (Figure 1) [51]. Both VEGFR and Tie2 are receptor tyrosine kinases (RTKs) which are highly expressed in EC whereas the Tie2 is also expressed in macrophages [52]. *Vegf* is transcriptionally targeted by HIF. The increased secretion of VEGF by macrophages activates VEGFR-2 on EC, inducing a robust tyrosine phosphorylation that enhances the cell migration, survival, and proliferation [27]. Moreover, activation of VEGFR-3 on endothelial tip cell by macrophage-expressing VEGF-C promotes the vascular sprouting and controls the fusion of vascular branches [53]. Therapeutic targeting the VEGFR with the overexpression of VEGF has been developed at an early stage for the treatment of PAD [54,55]. However, the clinical outcome of VEGF-A, either only or with other growth factors, is not satisfactory [56], which is attributed to the production of both pro-angiogenic VEGF_165_a and anti-angiogenic VEGF_165_b isoforms when spliced in exon-8 of VEGFA [57,58]. Apart from the VEGF-VEGFR ligand-receptor system, the Delta-like protein-Neurogenic locus notch homolog protein 1 (Dll-Notch1) is also involved [19]. Dll1 and Dll4 are the ligands of Notch receptors that are specifically expressed in ECs [20]. There is a feedback loop between the VEGF signaling and Notch signaling in EC. Specifically, VEGF pathway serves as an upstream regulator of Notch, which in turn suppresses VEGFR-2 expression during developmental angiogenesis [59,60,61]. Notch receptor expression is not restricted to ECs but also found on macrophages. In ischemic neovascularization, EC-expressing Dll1 is highly induced [20], which activates the Notch receptor on macrophages [19]. The activation of Notch signaling directs the differentiation of macrophages towards a phagocytic and anti-inflammatory phenotype, as a result promoting arteriogenesis and tissue repair [19].

The ANG-TIE system is defined as another essential regulator in embryonic vascular development and postnatal vascular homeostasis. ANG1 activates TIE2 signaling in EC to promote vascular remodeling and formation, while ANG2 serves as an antagonist of ANG1-TIE2 interaction [62]. The circulating ANG1 level is elevated in patients with critical limb ischemia [17], resulting in EC migration and proliferation. Apart from the expression of TIE2 on EC, the TIE2-expressing monocytes/macrophages (TEM) are recently identified to be pro-angiogenesis and important to revascularization in HLI. Activation of TEM by ANG1 induced the migration and M2-polarization of TEM [63]. Meanwhile, ANG1 can decrease the oxygen-sensitive prolyl hydroxylase domain protein 2 (PHD2) in macrophages, which in turn upregulate HIF-1α for inducing angiogenesis [16].

Apart from the involvement of HIF-mediating VEGF and ANG-TIE signaling, other proangiogenic factors are also reported to be secreted by macrophages, such as interleukin-8 (IL-8) [22], PR39 [26], matrix metalloproteinase-9 (MMP-9) [25], and CCL2 [22]. For instance, IL-8 is regarded as a key paracrine cytokine that regulates vascular permeability and angiogenesis. IL-8 stimulates the direct interaction of its receptor CXCR2 with VEGFR2 on EC, therefore inducing the transactivation of VEGFR2 [23], resulting in angiogenesis in ischemic retinopathy [22]. PR39 is a macrophage-derived peptide that suppresses the ubiquitin–proteasome-dependent degradation of HIF-1α. It is shown that the administration of PR39 improves the neovascularization in myocardial ischemia, indicating a therapeutic role of this peptide in tissue ischemic injury [26]. Although a variety of molecules are identified, the detailed communicational networks between macrophages and other cell types in injured muscle after ischemia are not fully understood. The chemokines, cytokines, ligands, and metabolites constituting the microenvironment would be promising therapeutic targets for ischemic injury.

### 2.5. Physical Interaction of Macrophages and Endothelial Cells Post-Injury

Vascular sprouting and fusion of tip cells initiate the repair of ruptured vessels. Self-proliferation of local microvascular ECs, and the recruitment and differentiation of bone marrow-derived endothelial progenitor cells both play important roles in angiogenesis. Apart from producing enzymes, growth factors, and cytokines to stimulate angiogenesis which have been discussed in Section 2.1, Section 2.2, Section 2.3 and Section 2.4, the physical interaction between macrophages and ECs (Figure 1) and its implication in injury and repair have also been studied by several groups taking the technical advantages of in vivo imaging techniques and tracing methods.

Early work investigating collateral proliferation using methods, such as fluorescent probe labelling and depletion of bone marrow cells, showed that homing blood mononuclear cells to the ischemic site plays a negligible role in collateral vessel growth, while CCL2 participates in the accumulation of tissue macrophages only locally to support vessel growth [64]. This finding highlighted the importance of local macrophages rather than macrophages derived from circulating monocytes to support arteriogenesis after injury. Later work established a direct interaction between macrophages and ECs using the zebrafish model and more advanced imaging methods. During the embryonic stage of vascular network development, macrophages interact with endothelial tip cells by bridge tip cells at the junctions in zebrafish [65]. In line with the previous work done in adult mice [64], although mediated by CSF1, monocyte-derived macrophages are not responsible for developmental angiogenesis [65].

In another study, newly formed vessels serve as a guide for Schwann cells to migrate toward nerve injury site through direct contact between new vessel and Schwann cells [49]. Macrophages located within these bridging points sense local hypoxia, produce VEGFA in a temporal manner in order to form a properly polarized vasculature which serves as a scaffold for Schwann cells to attach and regenerate axons [49]. A more recent study also supported the role of macrophages to respond to hypoxia and directly interact with ECs at various stages of vasculogenesis in zebrafish [48]. Apart from producing VEGFA to modulate the reconstruction of nerve-vessel bridge in response to ischemia, a later study showed that brain macrophages are also able to sense extracellular ATP which is commonly found at the site of vascular rupture, followed by reorganization and assembly of microfilament to form a physical adhesion directly to ECs, helping to repair vascular structure after cerebral vascular rupture in the zebrafish [66]. These findings highlighted the importance of macrophages to facilitate vascular repair and axon regrowth in brain injury.

Similarly, in the liver, monocyte-derived macrophages but not Kupffer cells can interact directly with ECs at the sites where VE-cadherin adherence is disrupted. Macrophages also deliver growth factors through these leaky points between themselves and ECs. In addition, the recruitment of monocytes is mediated by endothelial ICAM-1 during liver injury and regeneration [67]. Two notable features occur during these events. First, the direct contact of macrophages with other cell types involves easily distinguishable morphological changes for example extension of filopodia, possibly mediated by myosin IIA, as well as a partner molecule on the other cell type, e.g., VE-cadherin on ECs. Second, these macrophages could be highly dynamic, originated from both MoMac and TrMac. They can also sense various types of signals like hypoxia or ATP release from cellular damage depending on the context of the injury. However, whether such a relationship between macrophages with the endothelium, tissue-resident progenitor cells, and possibly parenchymal cells in other organs are also important for vascular injury and repair for example in the heart and skeletal muscles are not fully revealed.

## 3. Concluding Remarks

Many macrophage-secreted factors have been identified to be functionally important at various stages of vascular repair, yet not all have been tested in hindlimb ischemia model, especially under certain conditions together with common cardiovascular risk factors, such as high glucose, lipid dysregulation, aging, and smoking. Recent discoveries in the field showed some interesting cell-cell interaction, for example, between macrophages (either TrM or MoM) with stromal cells and parenchymal cells in various organs during the regeneration process, which need more research. The adequate and timely control of macrophage phenotypic switch during the whole process is necessary to maintain tissue homeostasis, as hyperactive inflammation exacerbates injury and dysregulated pro-resolving macrophages will lead to pathological fibrosis. How the phenotypic change of macrophages is precisely regulated at different stages is also the active area of research. To address these questions, animal models from zebrafish to rodents are widely used in preclinical studies. The accessibility of in vivo time-series imaging techniques in various animal disease models, with or without gene editing, would be of great advantage for mechanistic study. New techniques, such as single cell transcriptomics and high-dimensional multi-color cytometric analysis, can help to reveal a more detailed spatio-temporal involvement of macrophages at different stages of remodeling and repair, especially about the heterogeneity of macrophages and its interaction with other cell types. However, challenges also stand in the way. It is noteworthy to take the variability of animal genome and immunogenicity into consideration when dealing with research outcomes. Future studies could generate new insights into the development of new therapies.

## Figures and Tables

**Figure 1 ijms-21-06328-f001:**
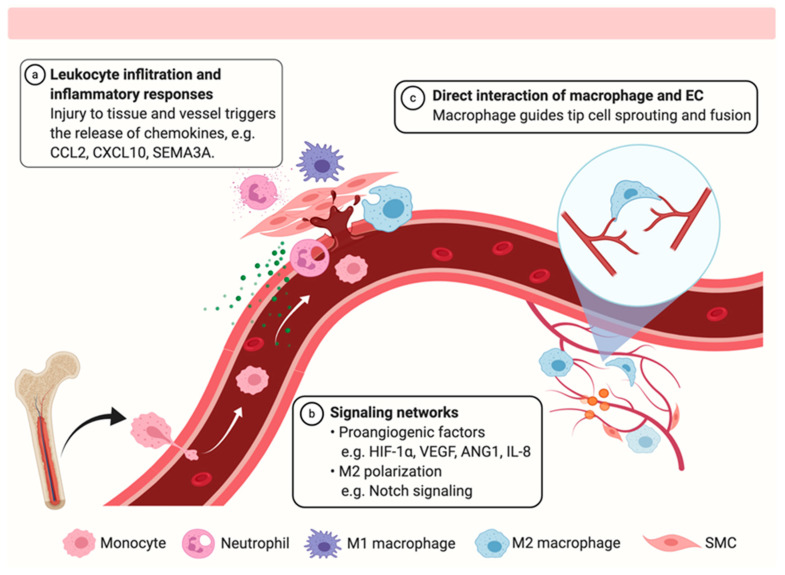
The role of macrophage during the vascular repair. Bone marrow-derived monocytes are recruited to the site of tissue injury in response to the chemokine gradient. Following the initial inflammatory responses, tissue-resident macrophages (TrM) and monocyte-derived macrophages (MoM) polarize to M2 phenotype for phagocytosis of necrotic cells and injured stimuli. During the proliferative phase, proangiogenic factors are induced to improve angiogenesis. Macrophages can bridge tip cells directly to guide the vessel sprouting and branching for neovascularization. The diagram is created with BioRender.com.

**Table 1 ijms-21-06328-t001:** Signaling molecules in macrophage-mediated vascular repair and neovascularization.

Functions	Signaling Molecules	Effects on Macrophages	Phenotypes upon Injury	Ref.
Chemotaxis and cell recruitment	AGE-RAGE	Reduce macrophage infiltration and interaction with EC	RAGE KO or overexpression of reduced AGE enhance vascular repair in diabetic mice upon HLI	[8]
	CCL2-CCR2	Recruitment of proangiogenic monocytes/macrophages	CCR2 KO impairs recovery of blood flow recovery, vessel size, and active foot movement in HLI mice	[9,10]
	CXCL10-CXCR3	Regulate leucocyte infiltration	CXCR3 KO reduces VEGF production, angiogenesis, blood perfusion, and capillary density	[11]
	PGC-1α-SPP1	Recruit macrophage and upregulate CCL2 production	PGC-1α overexpression improves angiogenesis and blood flow recovery in adult, aged, diabetic mice; SPP1 KO induces immature capillarization and blunted arterialization	[12]
	SEMA3A/VEGF-NRP-1	Recruit NRP-1+ macrophage	NRP-1 deficient macrophage fail to enter retinal and reduce neovascularization in OIR mice	[13]
	SERCA 2	Regulated VEGF production and adhesion to EC	Mediated immune cells infiltration and adhesion via ERO1 and VCAM-1 expression in EC	[14]
	VASP	Form complex with CCR2, suppress macrophage differentiation via STAT signaling	KO increase blood flow recovery, angiogenesis, arteriogenesis, and leukocyte infiltration upon HLI	[15]
Angiogenesis	ANG/TIE2	Upregulate HIF signaling via repressing Phd2 and M2 polarization	ANG or TIE2 overexpression increases vessel density, reduced ischemic necrosis in HLI mice	[16,17,18]
	DLL1-NOTCH	Promote differentiation from Ly6Chi monocyte, enhanced phagocytic capacity and anti-inflammatory phenotype	Heterozygous Dll1 mutant prevents arteriogenesis, blood perfusion, and tissue recovery in HLI mice	[19,20]
	HIF	HIF-1α KO reduced macrophage migration and suppressed pro-inflammatory phenotype	KO impairs ruptured vessel repair, angiogenesis, and tissue repair	[21]
	IL-8	M2 polarization	Blockade of IL-8 suppresses angiogenesis	[22,23]
	MiR93/IRF9/IRG1/itaconic acid	MiR93-mediated suppression of IRF9/IRG1/itaconic acid induces M2 polarization	MiR93 overexpression promotes angiogenesis, arteriogenesis, and blood perfusion	[24]
	MMP-9	Secreted by M2 macrophage	KO reduces capillary branching	[25]
	PR39 peptide	Inhibited the degradation of HIF-1 α	Promote angiogenesis	[26]
	VEGF-VEGFR	Activate NOTCH signaling, induce maturation and M2 polarization	Promote EC migration, proliferation and angiogenesis	[19,27]

Abbreviation in the table: HLI, hindlimb ischemia; OIR, oxygen-induced retinopathy.

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
