# Peer review of "The Role of Macrophages in Vascular Repair and Regeneration after Ischemic Injury"

_ijms, 2020, doi:10.3390/ijms21176328_

Round 1

Reviewer 1 Report

In the present manuscript, authors review some functions of macrophages in vascular homeostasis. Despite the fact that the topic is of the highest interest for readers, the review is of low quality and lack structure. Here is a list of comments that should help authors to improve their manuscript:

  • It will be more appropriated to discuss each function separately. For instance, the M1 polarization is an earlier event initiating the response while the M2 polarization is a later event. Instead of discussing all the polarization states together, it will be clearer to start with M1 and develop the role of M1-associated cytokines before talking about M2 polarization.
  • As a general comment, this review lists facts but does not explain or detail mechanism. Moreover, facts are provided all along the manuscript without linking or organizing them.
  • Macrophage function is restricted to the production of some cytokines and contact with endothelial cells, but there is much more that are not even mention in the manuscript such as phagocytosis, trans-differentiation, collagens production, NO, prostaglandins, proteolytic enzymes involved in ECM remodeling (MMP-9 is barely mentioned), …
  • Paragraph 2.3 « Signal induction of macrophages ». First, the title means nothing. Secondly, this paragraph mainly addresses endothelial cells which are outside the scope of this review.
  • English must be improved

Author Response

Response to Reviewer 1:
Comment: It will be more appropriated to discuss each function separately. For instance, the M1 polarization is an earlier event initiating the response while the M2 polarization is a later event. Instead of discussing all the polarization states together, it will be clearer to start with M1 and develop the role of M1-associated cytokines before talking about M2 polarization.

Response: We thank reviewer for the suggestion. We agree that the role of M1 and M2 macrophages have different functions at different stages during vascular repair. In the revised manuscript, we insert a summarized description of the role of M1 macrophages in section 2.1, before introducing the M2 polarization. Also, the M1-related cytokine IFNg-mediated phagocytosis is emphasized and discussed in section 2.3. However, we plan to focus more on the role of M2 at a later stage, so we did not expand the M1 part into more details, considering the scope of the review.

Comment: As a general comment, this review lists facts but does not explain or detail mechanism. Moreover, facts are provided all along the manuscript without linking or organizing them.
Response:  We thank the review for the comment. We include a summary of the common function of macrophages at the beginning of session 2. We also revised related parts by added more explanation and implication of mechanisms and interaction among mechanisms. With the schematic diagram, we hope to provide readers with a better understanding of the role of macrophages during vascular repair that facilitate the following discussion. 

Comment: Macrophage function is restricted to the production of some cytokines and contact with endothelial cells, but there is much more that are not even mention in the manuscript such as phagocytosis, trans-differentiation, collagens production, NO, prostaglandins, proteolytic enzymes involved in ECM remodeling (MMP-9 is barely mentioned), …

Response: We thank the reviewer for the constructive comments. We agree that the functions of macrophages are not restricted to what we mentioned in the manuscript. Indeed, the function of macrophages in tissue injury, such as skeletal muscle damage, myocardial infarction, have been revieweed in great details elsewhere, which includes the general role of macrophages such as pro-inflammation and ECM remodeling. In this review, we focused more on the involvement of macrophages particularly in vascular repair that is commonly associated with many cellular processes but not reviewed in the recent years. Therefore our discussion focused more on vessel repair, angiogenesis, and the cell functions involved in these processes, based on our search of literature. Although the functions mentioned by reviewer are also extremely important for tissue injury or repair, for example, ECM remodelling or transdifferentiation, they might not have been studied in details, and associated with macrophages in the context of vascular repair.

Comment: Paragraph 2.3 « Signal induction of macrophages ». First, the title means nothing. Secondly, this paragraph mainly addresses endothelial cells which are outside the scope of this review.

Response:  We thank the reviewer for pointing out this important issue. The title has been revised to "Pro-angiogenesis signaling networks between macrophages and endothelial cells”. The ligand-receptors system between macrophages and endothelial cells are interactive and important for both macrophages and endothelial cells function. As the endothelial cells are critical for vascular function, summarizing the communication between macrophages and endothelial cells during vascular repair would provide insights into the aspect of the microenvironment and cell-cell communication.

Comment: English must be improved
Response: The revised manuscript has been edited by native English speaking colleague and also scanned by the Grammarly (www.grammarly.com) to improve the language.

Reviewer 2 Report

This is an up to date review on the role of macrophages in vascular repair. Overall, it is well written, comprehensive and well referenced.
Issues to address
The main thing lacking form this review is an integrated synthesis of the various sections. For example, sections 2.3 and 2.4 describe chemical signals and physical interactions. However, it is likely that these different modes of signaling interact or amplify each other. Please elaborate on this theme.
Similarly, the last section on Concluding remarks needs to be expanded upon. Please address broad questions such as the following: What are the challenges most important to resolve on the role of macrophages in vascular repair; what are the methodologies most likely to be of benefit in addressing these problems; what contribution can be expected from current model systems, from zebra fish to Matrigel, and what are the limitations of these model systems
Minor issue
Although overall well written, the manuscript would benefit from English editing

Author Response

Response to Reviewer 2:
Comment: The main thing lacking form this review is an integrated synthesis of the various sections. For example, sections 2.3 and 2.4 describe chemical signals and physical interactions. However, it is likely that these different modes of signaling interact or amplify each other. Please elaborate on this theme.

Response: We thank the reviewer for the constructive suggestion. A summary of the general function of macrophages is added at the beginning of session 2 to facilitate the following expanding discussion. The regulatory role of macrophages in vascular repair is diverse in different stages so we review the recent findings in different aspects. The physical interaction between macrophages and tip cells is recently identified taking the advantage of in-vivo imaging technique, which is not fully understood. We hope this review providing information about the physical interaction would inspire further investigation on the detailed molecular mechanism. According to the reviewer’s suggestions, we have revised and added some interaction between different modes.

Comment: Similarly, the last section on Concluding remarks needs to be expanded upon. Please address broad questions such as the following: What are the challenges most important to resolve on the role of macrophages in vascular repair; what are the methodologies most likely to be of benefit in addressing these problems; what contribution can be expected from current model systems, from zebra fish to Matrigel, and what are the limitations of these model systems

Response: We thank the reviewer for the constructive suggestion. We expand the concluding remarks in the revised manuscript with a discussion about the challenges and the model systems.

Comment:
 Although overall well written, the manuscript would benefit from English editing

Response: We thank the reviewer for the suggestion. The revised manuscript has been edited by native English speaking colleague and also scanned by the Grammarly (www.grammarly.com) to improve the language.

Round 2

Reviewer 1 Report

The manuscript has been substantially improved. There is still some minor English editing to do but it can be achieved during the proofreading process.

Author Response

We thank the reviewer for the suggestions. We have gone through the manuscript carefully again and revised some grammar mistakes and improper usage. We hope that the language is now suitable for publication.
